# Defining a new perspective: Enterprise Information Governance

Alastair McCullough[1*]

[1] *Department of Computer Science, Oxford University, Oxford, United Kingdom*

### Abstract

This paper adduces a novel definition of regulatory enterprise information governance as a strategic framework that acts through control mechanisms designed to assure accountability in managing decision rights over information and data assets in organizations. This new pragmatic definition takes the perspectives of both the practitioner and of the scholar. It builds upon earlier definitions to take a novel and more clearly regulatory approach and to synthesize a new definition for such governance; to build out a view of it as a scalable regulatory framework for large or complex organizations that sees governance from this new perspective as a business architecture or target operating model in this increasingly critical domain. The paper supports and enables scholarly consideration and further research. It looks at definitions of information and data; of strategy in relation to information and data; of data management; of enterprise architecture; of governance, and governance as a type of strategic endeavor, and of the nature of strategic and tactical policies and standards that form the basis for such governance.

### Keywords

Data Governance, Information Governance, Enterprise Architecture, Enterprise Information Governance, Data Management, Data Product, Data Object, Policies, Standards, Regulatory Framework, Data Strategy, Target Business Architecture, Operating Model.

## 1. Introduction

Looking across the scope of what scholars and practitioners across the literature today call "Data Governance"[1, 2], there is a pressing need to re-frame and define the domain pragmatically in a knowledge space where today Oxford's Bodleian Library includes very well over two thousand references to titles containing the term. Definitions of such governance in the literature see information governance neither as a business architecture nor as an operating model: This paper takes the perspectives of both the practitioner and of the scholar to do so in the context of enterprise architectures and to offer a different perspective to reframe such governance to enable data strategists, governance designers, enterprise, business and data architects, scholars and researchers to use this clearer definition as a basis for their work.

The paper builds upon earlier researchers' definitions to take a novel and more clearly regulatory approach and to synthesize the new definition of such governance; to build out a view of it as a scalable regulatory framework for large or complex organizations that sees information governance as a business architecture. This view includes data strategy and

*NXDG, NeXt-generation Data Governance workshop, September 17, 2024, Amsterdam, Netherlands*

[*] Corresponding author.

✉ alastair.mccullough@cs.ox.ac.uk (A. McCullough)

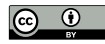 0009-0005-1324-1975 (A. McCullough)

enterprise architecture perspectives, a considered definition of information as a superset of data in this context, and the differentiation between the management and the governance of data. It is specifically framed both for field practitioners and scholars reviewing, designing, assessing, and researching governance and the target business architectures or operating models of governance. The paper supports and enables scholarly consideration and more advanced research and thinking in this increasingly critical domain.

The paper explores and proposes a specific, re-framed term and new definition, "Enterprise Information Governance," to characterize a regulatory framework in this space as a strategic approach, or meta-regulation. The paper takes this new approach by using the term "enterprise" in the same way as enterprise architects do: Industry analyst, Gartner®, defines enterprise architecture as "a discipline for proactively and holistically leading enterprise responses to disruptive forces by identifying and analyzing the execution of change toward desired business vision and outcomes."[3]

First, information and data are considered as concepts, then data strategy, and thinking around this is reviewed. The paper then looks at information and data governance, what it might mean; the differentiation between management versus governance; and adduces a novel definition for enterprise. It will use this also to discuss regulation of data via policies, standards, and procedures as part of governance. The paper uses a graphic representation to enable the reader to understand the new definition with clarity.

## 2. Information, Data, Big Data, and Data Objects

Information and data are as essential to organizations today as they have ever been, but their situation and disposition are far more complex today than they have ever been too, not least with increasing and popular interest in Artificial Intelligence dominating academic and commercial thought. One term that has become hackneyed with over-quotation is "big data". We can characterize it as data with volume, variety, velocity, and veracity requiring solutions that do not natively fit with traditional approaches to handling information management challenges.[4] Yet in a business and consumer world in which we see IT industry-dominating platform service providers, and fast-evolving AI technologies, landscapes, and ecosystems, finding a path forward in terms of governing such big data is far from simple.

Data scientist Clive Humby's famous and popular observation in 2006 that "data is the new oil"[5] has become a truism. Yet, sixteen years later, Forbes magazine's Nisha Talagala recently observed, "it is not enough to have data. One needs to have a Data Practice – a commonly understood and consistently executed set of principles for managing data."[6]

An abiding question in information technology today revolves around how to achieve the proper management of three traditional areas that consultants consider when they work with their clients in the technology domain: People, process, and technology.[7] Some form of "governance" is the solution for which leaders search and the terms "data governance" and "information governance" are widely used. We can find them today appearing internationally in industry, commerce, government and increasingly in academia, too. Considering Talagala's theme of principles for managing data, to align people, process, and technology with respect to data, this paper works towards a clearer understanding of what it is that needs to be governed.

There is an unresolved tension hidden in governance texts, and the research and the design of governances. It lies in the use of the terms "information governance" and "data

governance". Partly this is because there are common understandings used in the IT industry, and to which people come by diverse means. Partly, the tension arises because of uncertainty about what each term might mean: They are functionally woolly. But partly, too, with an emergent discipline, the meaning of governance itself can be ill-defined and misunderstood. This paper looks here at the terms "information" and "data", to provide a baseline for deeper, and different, definition of governance.

James Luisi sees the role of enterprise architecture as lying in the development of "various frameworks for IT that both facilitate the direction of the business and address major pain points... thereby aligning business vision and strategy with IT delivery"[8] and its mission as developing "the skills, business principles, standards, and frameworks for each architectural discipline necessary to prepare the IT resources of the organization to act effectively to achieve the direction set by the executive leadership team."

Steve Lockwood and their co-authors define enterprise architecture as providing a "framework for the business to add new applications, infrastructure, and systems for managing the lifecycle and the value of current and future environments,"[9] determining that it, "provides the alignment across business strategy, IT strategy, and IT implementation. It tightly integrates the business and IT strategies to create an ongoing way to use IT to sustain and grow the business."

Paul Brous, Marijn Janssen, and Riika Vilminko-Heikkinen say that "There is much confusion about what 'data' really is. Data is a set of characters, which have no meaning unless seen in the context of usage. The context and the usage provide a meaning to the data that constitute information."[10] "Data" are typically held in operational data stores and varietal components that constitute information technology systems and "systems of systems."[11] Increasingly, such data are processed at scale [12] to address a variety of requirements, and concerning a wide variety of uses.

Boisot and Canals [13] argue that "the difference between data, information, and knowledge is... crucial." They summarize, "information is an extraction from data that... has a capacity to perform useful work."[14] Also, that "The utility of data resides in the fact that it can carry information about the physical world; that of information, in the fact that it can modify an expectation or a state of knowledge." Their graphical representation of this [15] and commentary show they view data as effectively an informative lower, or more granular level, of factual insight or knowledge input, subsumed by information. The authors quote Roland Omnès in observing that, "data are for us a macroscopic classical fact... The datum is an essential intermediary for reaching a result."[16]

Considering the life of these data, they can be said typically to follow what might be called "a lifecycle"[17] in context in an organization. Such a lifecycle can be seen as beginning with data creation or collection, and transitioning through their processing, dissemination, use, storage in operational data stores such as databases and data marts, and disposition (for example, in an archive of less frequently used information), through to their destruction and deletion when no longer needed, or for security, regulatory or legal reasons. Part of the essential management thinking around data is the concept of treating them as intangible assets [18]. That is, broadly speaking, things that contribute to final production in an economy [19].

Weber, Otto and Österle observe that, "Data is often distinguished from information by referring to data as 'raw' or simple facts and to information as data put in a context or data that has been processed." They are satisfied that the terms data and information can be used

interchangeably [20]. Humby noted in their 'New Oil' speech that, "It is easy to grab a single fact and extrapolate that one fact into an actionable direction. But without context a fact is just that: A fact. It is not an insight."[21]

Michael Buckland [22] introduces the concept of "Information as Thing," in respect of which they determine that "information" can take forms as a process, as knowledge, and ("as thing") attributively for objects such as data and documents that are informative. Information is, by implication, a superset of the facts represented by data. Buckland sees "data" as implying "the sort of information-as-thing that has been processed in some way for use", and as denoting computer-stored records.

Michael Madison [23] distinguishes Data-as-form ("data seem thing-like... capable of exclusive ownership and control and subject to regulation as if [they] were an artifact") and data-as-flow ("wave-like, fluid, continuously evolving, even moving, aggregations of information.") In considering the lifecycle of data, essential to information architectures, through which data flow and in which they are held and stored; also in the lifecycle of data as they pass around an ecosystem. Madison notes, "The key point, illustrated by the necessity of metaphor, is that data are simultaneously form and flow."[24]

In consideration of the regulation and control of data, both their fluidity, and "thingness", will require regulatory governance as well. A useful perspective considering the oil analogy, Madison determines "'Data as [the new] oil' can be misleading. Oil is tangible, and oil reserves are depletable. In most senses, data are intangible, and pools or collections of data are not depletable"[25]; but also, "Data-as-flow captures the metaphorical instinct to look at data's fluid attributes."[26]

It could be determined from these analyses that "information" might be viewed as a superset in a technology landscape, and "data" a subset, in a consideration of semantics, information versus data. This construct would be useful, in that it would enable a consideration of information and data as terms at relative levels within and across an enterprise. "Information" can then be seen as an overarching term, more strategic and encompassing; more aligned with (and align-able to) a defined strategy and the IT context.

"Data" could be seen usefully as more operational, processable, resources: Things – assets- to be governed and controlled at a lower level, perhaps; and more associated with delivery, with software interaction and day-to-day activities in departments and teams.

In a 2021 paper, IBM® conceived of a particular concept, used in the design of governances for consulting clients, that of the "data object". It was defined as: "the information or data owned within the scope of ownership."[27] It introduced concepts of ownership of data, of their stewardship and custodianship, but also of a bounding of ownership, a domain or scope. IBM said that data objects must be: "Owned by a named Data Owner; Stewarded by a named Data Steward; In the Custody of a named Data Custodian."[28] Data owners "make decisions about the data to address the needs of their business function or the wider organization." Data stewards "are managers of the data (information) and of its implications." Data custodians "work closely with Data Owners, Data Stewards, and data security and protection (CISO) teams to define data security and access procedures, administer access systems, manage the disposition of the data day-to-day."[29, 30] A data object may physically relate to one or more data items. It can be constituted from a single byte, or many petabytes or zettabytes of data: "There is no physical or logical limit to the size, volume, or scale, of a data object in terms of

governance. The storage mechanism with which or upon which a data object is stored is not relevant to its governance, but may be relevant to its ownership."[31]

This conceptualization recalls Buckland's "information-as-thing" and implies clearly that data have been processed in some way, too. But this also resonates with Madison's concept of data-as-form. The concept of there being no limit to volume, to scale or size is pertinent to a consideration of "big data."

Synthesizing from Humby's idea about context, Gartner, Luisi, Boisot and Canals, Weber, Otto and Österle, Omnes and Buckland, and Madison, there appears to be clearly emerging a concept of "enterprise information." In fact, this is already encapsulated by Lockwood, et al., as, "Enterprises need to achieve information agility, leveraging trusted information as a strategic asset for sustained competitive advantage… Companies need to have… a comprehensive, enterprise-wide approach for information strategy and planning, and it is this which is considered an enterprise information strategy."[32]

## 3. Strategy and Data Strategy, Vision, and Business Architecture

In this paper's discussion of the governance of information and data, both the consideration of strategy and of tactics are of significant moment. In theory, it might be determined that a strategy is about exploring and defining activities [33]: It should explore the path to be taken, formulate an approach, and then determine the initiatives and activities out of which that approach should be composed across time, for example one-, three- and seven-year time horizons [34]. And it should be strategic, per se, not tactical. With the knowledge of a defined strategy before them, leaders can then take decisions about which funding tranches they need, which initiatives to support, when and why, with clear and coherent reasoning and understanding. In thinking about governance, this approach is deserving of considerable merit, not least because such a strategy determines a defined plan of action; a formulation leading us towards Nisha Talagala's "Data Practice – a commonly understood and consistently executed set of principles for managing data."

The Oxford English Dictionary notes that the noun, "strategy" is derived from the ancient Greek, στρατηγία ("strategia"), meaning "office or command of a general, generalship," though perhaps the definition that might be most accurate in an organizational context here is the wider and more holistic, "The art or practice of planning the future direction or outcome of something; the formulation or implementation of a plan, scheme, or course of action, esp. of a long-term or ambitious nature. Also: Policy or means of achieving objectives within a specified field, as political strategy, corporate strategy, etc."[35]

Michael Porter sees the essence of strategy in choosing unique and valuable positions that are rooted in systems of activities [36]. Focusing from generic strategy more towards information technology, Phillip Ein-Dor and Eli Segev [37] argue that strategic planning in an information systems context is about choosing objectives and deciding in what way they should be achieved.

Richard Wrangham sees a distinction between strategy and tactics: "…strategy was the art of the commander-in-chief 'projecting and directing the larger military movements and operations of a campaign,' while tactics was 'the art of handling forces in battle or in the immediate presence of the enemy.'" [38] Lawrence Freedman determines that strategy "is about getting more out of a situation than the starting balance of power would suggest. It is the art

of creating power,"[39] which draws together some of the implications of our previous definitions and goes further to add the nuance of organizational politics, and an implication, perhaps, of something of the order of leadership and maybe ownership. Freedman later adds a caveat to his definition to the effect that it allows the "impact of strategy to be measured as the outcome anticipated by reference to the prevailing balance of power and the actual outcome after the application of strategy."

Adding now, in this context, the term "data" to "strategy", relating Humby's new oil with the consistently executed set of principles for managing data, then technology industry analyst Gartner defines a "Data Strategy" as "a highly dynamic process employed to support the acquisition, organization, analysis, and delivery of data in support of business objectives."[40]

Weber, Otto and Österle [41] see the "strategy perspective" as including a corporate data quality strategy and strategic objectives, the business case for corporate data management (including a "status quo" maturity assessment), and a "portfolio of data quality initiatives" which they align with the governance of data. They also link the concepts of strategic direction, advocacy, and sponsorship into their view of an essential executive role ("Executive Sponsor") that provides oversight, and that we might interpret as embracing that "art or practice of planning the future direction" we met in our definition of strategy and the "balance of power" of Freedman, and that we might suggest brings together leadership and ownership, too. Lockwood, et al., note that an enterprise information strategy will establish principles to guide an organization's working towards an enterprise information architecture and will also provide "an end-to-end vision for all aspects of the information."[42]

Jack Welch framed the concept of "vision" in an interview with Harvard Business Review: "Good business leaders create a vision, articulate the vision, passionately own the vision, and relentlessly drive it to completion."[43] The framing of vision in this articulated way can drive governance coherently: Paul Brous, et al., tie neatly together strategy and the governance of data: "Governing data also includes ensuring compliance to the strategic, tactical and operational policies which the data management organization needs to follow."[44]

The end-to-end vision of Lockwood, et al., acts as a framing for Brous' tactical operational policies, for management. Welch's created, articulated, and owned vision acts as a driver for action. Freedman's concept of applied strategy sees the creation of power and of outcomes that are measurable.

In 2018, IBM defined an approach for information and data strategy that saw a "vision statement"[45] as setting out the current and future view and as being enterprise-wide or global in scope. A "mission statement" definition then set out how to provision that vision: Mission was seen as more localized to operational aspects or to specific localities, jurisdictions, or operating units. The use of mission statements here is validated by Alegre, et al., who conduct a systematic literature review of them, considering their pervasive use across multiple organizations.[46]

The IBM view was of "Vision, Mission, Goals" ("VMG") as a framing construct for leading governance strategically, and in working with businesses. At the lowest operational level, the defined goals related to the mission in the context of vision and set out the operational detail to deliver mission against vision. Goal statements and components such as initiatives to be realized via projects, could include a roadmap plotted against time. This approach aligns with Ein-Dor and Segev's [47] strategic planning and objectives; also, Weber, Otto and Österle's

[48] objectives, and Paul Brous' framing of policy, tactics, and management. [48] In 2024, IBM said that, "With any good data strategy, buy-in matters. To align business and data priorities, you need a clear understanding of the aims of the organization and senior leadership."[49]

Metadata software and services company Atlan™ include the concept of policies in relation to vision. They see data governance policy as documenting "the vision for data governance" and going further, "to list the actionable steps, and do's and don'ts imperative to realize that vision." They give three examples of policies such as a data usage policy; a data access policy; and a data integrity and integration policy. As examples of (external) regulations pertinent to policy, they list GDPR, the EU's General Data Protection Regulation 2018; and HIPAA, the US' Health Insurance Portability and Accountability Act 1996. Atlan feel it important also to "include guidelines for ensuring that an organization's data and information assets are managed consistently and used properly."[50] Guidelines are likely in this context to be either training and education artefacts that assist the activities of governance, or concise statements issued by an authoritative body that explore how a regulation such as a policy or standard should apply to a situation, technology or operation.

Looking at the modern contexts of information and data, their governance and strategy, Hanisch, et al., find "The strategic relevance of governance results from its ability to ensure and enhance performance... governance serves as not only a performance enabler but also a strategic differentiator."[51] The authors note that "The governance challenge involves creating mechanisms that help integrate, direct, and monitor ...distributed efforts," and that, "...governance broadly concerns the establishment of rules that help verify inputs and outputs (i.e., control mechanisms), divide and allocate tasks (i.e., coordination mechanisms), align competing interests (i.e., incentive mechanisms), and attenuate relational vulnerabilities (i.e., trust mechanisms)."[52]

In a consideration of strategy, of the design of governance, establishment of rules, division and allocation of tasks, alignment of competing interests, and technology competence, there is a highly useful approach, a definition, or blueprint that is used by business architects [53] to frame the necessary design. This is known as an "operating model", "target operating model"[54, 55] (also known as a "TOM") or as a "target business architecture."[56] The TOM can be defined by a competent practitioner to specify both strategic governance and tactical governance, relevant to the specific technology domain. A TOM is itself a strategic artefact, part of a data strategy's design and a tool that can be used to bring governance into the organization in a concrete, actionable form. It can be socialized with stakeholders and sponsors, referenced, and maintained, published and promulgated. Such information and data governance design could also be determined to be a style or method of business architecture. Hadaya and Gagnon note that a "...target business architecture makes it possible to determine the business capabilities, functions, processes, organizational units, knowledge, information and branding that the organization will need. It defines also the main characteristics of these elements and their desired interrelationships."[57]

Summarizing, research shows that information and data governance is itself a form of data strategy. It broadly concerns the establishment of control mechanisms, coordination mechanisms, incentive mechanisms. It aims to ensure compliance with strategic, tactical, and operational policies. We can relate it to the strategic vision, mission, and goals of an organization. Vision, here, is a driver for action, and acts as a framing for tactical operational policies, management, and of measurable outcomes. The framing of the design of such

governance can be seen as a business architecture, target operating model, or "TOM." This is a new perspective in the framing of such governance by comparison with existing literature in the field.

## 4. Data Management and Data Governance

The concept of valuing data leads to a consideration that they might be managed and governed in a consistent and coherent manner as well, and we have just considered how strategy and governance align. A survey of the data and information governance domain shows that the terms "data governance" and "information governance" based upon a study of authors who have considered a very wide range of sources, yields no apparent, single and agreed canonical definition, though scholars have worked hard to determine one [58, 59, 60]. Abraham, Schneider and vom Brocke summarise the situation in their structured review that, "We did not find a standard definition of data governance in scholarly literature or in the set of practitioner publications."[61] However, the varietal definitions offered enable us in practice to understand governance' components very usefully and with some considerable refinement, and to work towards a novel candidate definition, which we shall determine, to support consideration in this paper.

By way of reviewing etymology for a moment, the term "governance" is derived from Anglo-Norman, "governaunce" or "gouvernaunce". The OED tells us that it means, "The office, function, or power of governing; authority or permission to govern."[62]

Madison sees that "with respect to data, we should be asking about governance, not asking simply about law," [63] and differentiates data governance from jurisprudential law, its regulation and public policy. Further, "The concept of governance is used here in the sense of collective or coordinated decision making by individuals working together, about decisions on matters of collective interest"[64] and that "data governance is above all else, perhaps, a complex and sustained challenge in managing shared resources in institutional contexts."[65]

Susan deMaine [66] suggests that "Information governance is a holistic business approach to managing and using information that recognizes information as an asset as well as a potential source of risk." deMaine says that, "The term 'data governance' is also evident in the literature. Sometimes this term is used to mean essentially the same thing as information governance. At other times, the term is used more narrowly, focusing on the nature and integrity of the data artifacts themselves rather than the knowledge they represent."[67]

Abraham, Schneider and vom Brocke see the purpose of such governance as, "The exercise of authority and control over the management of data... to increase the value of data and minimize data-related cost and risk."[68] It would seem, now, apt that this concept of "value" should be added to the big data characteristics from our introduction, so, volume, variety, velocity, veracity, and value: "Five Vs."

Industry body, the Enterprise Data Management (EDM) Council determine data governance to be, "The function that defines and implements the standards, controls and best practices of the data management initiative in alignment with strategy," and that it "...is responsible for creating and implementing a data control environment."[69]

Inge Graef, working in the context both of information governance and data protection in their editorial piece, and incidentally considering the structures of governance and forms of

control over data, sees information governance as "including legislative and regulatory actions to enhance the creation of value from data."[70]

Robert Seiner, tending to align with Abraham, Schneider and vom Brocke, sees data governance as "the formal execution and enforcement of authority over the management of data and data-related assets."[71]

Olivia Benfeldt Nielsen prefers to use a term ("framework") in seeing the artefacts and materials of governance as, "a framework for decision rights and accountabilities to encourage desirable behavior in the use of data"[72]. Benfeldt Nielsen, further, quotes both Pierce, et al. as defining such governance as "the collective set of decision-making processes for the use and value-maximization of an organization's data assets,"[73] and Otto as using the same, framework terminology as Benfeldt Nielsen in defining it as, "a companywide framework for assigning decision-related rights and duties in order to be able to adequately handle data as a company asset."[74] Separately in their own paper, Weber, Otto and Österle note that, "data governance specifies the framework for decision rights and accountabilities to encourage desirable behavior in the use of data."[75] Further, that, "To promote desirable behavior, data governance develops and implements corporate-wide data policies, guidelines, and standards that are consistent with the organization's mission, strategy, values, norms, and culture."[76]

Robert Smallwood differentiates between more traditional information technology governance, which he finds "consists of following established frameworks and best practices to gain the most leverage and benefit out of IT investments and support accomplishment of business objectives,"[77] and data governance, which he sees as "the execution and enforcement of authority over the definition, production, and usage of data,"[78] which results in increased trust and accuracy in data used by an organization. Smallwood states that it "consists of the overarching polices and processes to optimize and leverage information", and "processes, methods, and techniques to ensure that data at the root level is of high quality, reliable, and unique (not duplicated)."[79] He expects that governance will control the access to information, assure its security and meet a range of obligations, regulatory, legal and privacy. Smallwood observes what he feels are important features, too: Governance in his view is a "multi-disciplinary program that requires ongoing effort."[80] So, not merely a short-term project or programme of work; and he feels it should, "focus on breaking down traditional functional group 'siloed' approaches."[81] In respect of research here, we can see this chimes well with the finding of Rene Abraham, et al., that: "Data governance specifies a cross-functional framework for managing data as a strategic enterprise asset."[82]

Boris Otto, in his 2011 paper on the morphology of data governance organizations, brings out the idea of location and scope, writing of the concept of "locus of control"[83] as "the main instance of responsibility for data governance in a company." Otto brings out the variety of different authors' views with respect to the "hierarchical positioning" of the locus, for example in different functional business departments versus IT/IS department, versus a shared responsibility. Otto notes that there is no clear trend across differing opinions and observes that centralized and decentralized organization is effectively a continuum.

Vijay Khatri and Carol Brown's research develops a "framework for data decision domains" and what they define as a "framework for data governance" also determines the concept of a "locus of accountability." In our paper here, we can set this as against, or as an adjunct to, Otto's "locus of control" in considering management and governance concepts.

Khatri and Brown see governance in the context of *IT governance* and *information and IT* as asset; in this view of what this paper's research has seen as information-as-asset, and therefore effectively aligning with Lockwood, et al., Pierce, et al., deMaine, Seiner, Atlan, and by implication, Madison, and Abraham, Schneider and vom Brocke too. Khatri and Brown determine that "information assets (or data) are defined as facts having value or potential value that are documented."[84] They differentiate between data governance and data management: "Governance refers to what decisions must be made to ensure effective management and use of IT (decision domains) and who makes the decisions (locus of accountability for decision making). Management involves making and implementing decisions."[85]. The positioning of Khatri and Brown's locus of accountability depends upon the way in which the operating model of the organization supports (or fails to support) governance of enterprise information and its ownership. Their definition is also dominated by "data governance", which we have determined earlier in this paper, differentiates from information governance by virtue of being a subset, or lower level, of it. They determine the concept of "Data Principles", which "establish the extent to which data is an enterprise wide asset, and thus what specific policies, standards and guidelines are appropriate" [86].

In a joint paper by the British Academy and the Royal Society in June 2017, working group members considered that data governance means, "everything designed to inform the extent of confidence in data management, data use and the technologies derived from it."[87] Their paper includes the... "...institutional configuration of legal, ethical, professional and behavioral norms of conduct, conventions and practices that, taken together, govern the collection, storage, use and transfer of data and the institutional mechanisms by and through which those norms are established and enforced."[88]. It sees the management of data and the use of data as inseparable and, aligning with Benfeldt Neilsen, Otto, Weber, Otto and Österle, and Rene Abraham, et al., refers to the concept of a "governance framework... to ensure trustworthiness and trust in the management and use of data as a whole."[89]

The management of data typically requires a set of software products, tools and techniques that are clearly differentiated from the regulatory governance of data [90, 91], the focus of this paper. Research up to this point shows evidence indicating information and data governance is implicitly a meta-set of data management functionally and differentiated from it: Such governance needs to apply to, regulate, bound and scope activities with respect to, and provide rules in relation to the proper operation of software, tooling and methods of management of data. Governance cannot, therefore, be synonymous with them operationally if it is itself a meta-activity.

There is a high degree of practicality here, not least in thinking about the reality of our "Five Vs" in the delivery of outcomes –investments, projects, programmes, governances, tactical and strategic activity- in real world organizations. In fact, data management will typically have an operational escalation path, particularly in relation to service level (SLA)[92] (also discussed by Khatri and Brown) and operational level (OLA)[93] agreements against which services providing data to customers, internal or external to the organization, are effected. Escalations are defined typically for service levels that relate to the way that data management issues are resolved. For example, a "Sev 1" (Severity One – critical, with high impact) level escalation might be directed for resolution to a supplier company, whilst a "Sev 3" (Severity Three –minor, with low impact) level might be managed internally by a functional service support team [94]. Unlike service management or data management, data and information governance issues and

escalations will typically escalate up a path to differentiated governance bodies such as the Information Management Office (IMO) or Enterprise Data Office (EDO), Data Strategy Board (DSB) or Information Governance Council (IGC).[95]

As a result, the escalation paths will not necessarily be equated, though they may be parallel: Staffing differs; policy and guidelines differ. Resolutions will have differing categories of outcomes in terms of personnel, technical activities, and actions between governance versus management of data and information. There is an alignment in thinking here between Smallwood's concept of information technology governance, Weber, et al's, concepts of governance developing and implementing corporate-wide policies, and Seiner's and Benfeldt Nielsen's research findings.

Alhassan, Sammon and Daly neatly sum up research in this domain by noting that the terms 'governance' and 'management' differ in their view because, "...governance refers to the decisions that must be made and who makes these decisions to ensure effective management and use of resources, whereas management involves implementing decisions. Hence, management is influenced by governance."[96]

We have referred to "standard" or "standards" seven times already, but not really defined the term in this context. Looking across the literature in the data governance domain shows that there are few clear and actionable definitions of the term as it might be used in actual relation to the regulatory governance of information and data. Authors seem to tend to use the word as part of a list of typical artefacts they might expect to see produced in relation to governance activities with an assumption that the reader will understand implicitly what they might mean. Smallwood [97] turns to more legalistic definitions, preferring to refer to "De jure ('the law') standards... published by recognized standards-setting bodies" and "De facto ('the fact') standards" that are "not formal standards but are regarded by many as if they were," and gives the example of some International Standards Organization (ISO) standards that are more of the order of technical reports, but have no legal enforcement aspect. Gartner [98] offer, "A document that recommends a protocol, interface, type of wiring, or some other aspect of a system," and say that "De facto standards are widely used vendor-developed protocols or architectures." But they also say, confusingly, that "Standards" can be defined as, "Specifications or styles that are widely accepted by users and adopted by several vendors."[99]

A definition of standards that seems particularly germane to this paper is one provided by Janet Lichtenberger [100], who sees them as "the precise criteria, specifications, and rules for the definition, creation, storage and usage of data within an enterprise." Standards, in governance context, could include those for naming conventions, for quality measures, retention rules, and backup frequencies. The relevance of Lichtenberger's presentation is interesting: It is hosted online by the Minnesota chapter of DAMA, the internationally known Data Management Association, so it would be fair for a researcher to assess such an august body as satisfied with its relevance and value.[101]

Lichtenberger defines policies in governance context, though uses "data management" (rather than governance) as a generic term: "...the overall business rules and processes that an enterprise utilizes to provide guidance for data management. Policies might include adherence of data to business rules, providing guidance for protection of data assets, compliance with laws and regulations, defining enterprise data management functions, and others."[102]

With real-world governance there is an apparent overlap between data management, data security and data privacy. In their paper on cloud data governance, Al-Ruithe, Benkhelifa

and Khawar say that "Data governance issues for concern include risk management, disaster recovery plan, security, privacy, integrity, incident response, access management, and accountability."[103]   From considering literature in the domain, it appears fairly straightforward to understand that where regulatory governance will need to encompass more technical topics, including aspects of personal data management, data security, data privacy and topics that might otherwise be seen operationally most appropriately (for example) in the sphere of the Chief Information Security Officer (CISO), Data Protection Officer (DPO) or Chief Privacy Officer (CPO)[104], policies and standards could be defined that specify the relevant regulations appropriately and such that the overall governance can encompass these domains effectively.   Data software specialist company Informatica say that "Business policies and standards are critical for any data governance program," [105]  and give example policies as including those with relation to data accountability, ownership (Khatri and Brown's,  term "data trustees", appears analogous), organizational roles and responsibilities, data capture and validation, security and data privacy, data access, data usage, data retention and data archiving.[106]  These considerations echo the big data definitions of veracity and value.

Summarizing, "policies" represent the way in which data should be governed in the context of a particular organization: Policies define the relative vision for data governance (and that could align with IBM's concept of VMG), actionable steps, the shape of governance; and "do's and don'ts" that are imperative to realization of the vision.  Examples of policies can include data usage, data access, and data integrity and integration.

"Standards", differentiated from policies, describe the parameters which apply to data and that are referenced in the related policy or policies.  They represent the finer grained detail in terms of delivery.  A policy, therefore, may have no standards, one standard or many standards that implement it.

Just as there may be a policy for data access, there may be a "Data Access Standard" that explores the parameters and actionable regulatory components –criteria, specifications, and rules for the definition, creation, storage and usage of data- for which the (top level) "Data Access Policy" conveys the overall rules and processes, adherence of data to business rules, guidance for protection, compliance with laws and regulations, and definition of enterprise data management functions related to data access.

We noted earlier that the management of data is typified by a set of software products, tools, and techniques.  Such management could be clarified by thinking of it more exactly as the work related directly to operations upon and with data –for example, their access, ingestion, extraction, transformation, loading, movement, storage, master (MDM), meta- and reference (RDM) data management, visualization, quality assurance, security, archiving- and the "monitoring and auditing of the way data are used within the organization to ensure [they are] aligned with the governance that has been emplaced, including by analyzing the quality and reviewing the content," ultimately for the organization to "enable quality data with trusted provenance,"[107] and thinking back to our Five Vs' veracity and value.  We can see here more clearly the difference between "decision rights and accountabilities"[108] and such management of data.  We can also recall Smallwood's observation that governance should "focus on breaking down traditional functional group 'siloed' approaches;" [109] Rene Abraham, *et al's,* "Data governance specifies a cross-functional framework for managing data as a strategic enterprise asset;'" [110] and DeMaine's, "Top-down implementation of information governance is particularly effective at taking the holistic view of information" [111].  Khatri and Brown quote

Carol Brown's 1999 paper in observing that, "In designing data governance, the assignment of the locus of accountability for each decision domain will be somewhere on a continuum between centralized and decentralized." [112]

## 5. Synthesizing a novel "Enterprise Information Governance"

Taking the concepts of management and governance as differentiated, and synthesizing across writers and scholars considered in this paper, a candidate, and novel, definition for the governance domain is now feasible. This can be used to support data strategists, governance designers, enterprise and business architects, scholars and researchers in using a novel definition as a basis for their work and in designing operating models specific to enterprise.

Information and data governance, can be represented as a novel representation as "enterprise information governance" [113, 114]. It encompasses both information governance and data governance [115, 116, 117, 118, 119]. Enterprise Information Governance is a corporate-wide strategic [120, 121, 122] framework [123, 124, 125, 126, 127], differentiated from pure information technology governance [128] and data management [129], and defined against a vision for end-to-end governance [130, 131, 132]. It frames actions that ensure trust and compliance with strategic, tactical, and operational policies, standards, guidelines, processes [133, 134, 135, 136, 137, 138, 139, 140]. The framework is intended to support the management of shared resources [141], and is delivered through rules or control mechanisms [142] that help to co-ordinate, integrate, direct, monitor and allocate tasks [143], exercising authority, control, and accountability in managing decision rights over data [144, 145, 146, 147, 148]. Enterprise information governance is undertaken in pursuit of value-maximization of an organization's data assets [149, 150] and consistent with its strategy, mission, values, norms, and culture [151, 152]. A framework that represents the governance may now be defined using a target operating model or business architecture to bridge the gap between strategic vision and tactical day-to-day operations [153, 154], and aspects of its design lie within the realm of business architecture [155, 156].

Valid as this paragraph might be, it is complex to understand textually. A novel graphical representation to support visualization of this new synthesis provides a simpler view and **Figure 1** represents this, derived from this paper's synthetic research work with relative levels of components determined by reference to contexts of source research, for example with information as superset of data; strategy and mission as overarching drivers for action – strategy ideally serving as a framing for vision, mission and goals. In the graphic, the Information Technology Operating model component, adduced from sources discussing IT governance, is lacking detail because it is mainly outside the scope of this paper. The framework graphic here is offered as a way for the reader to visualize this novel framing of the definition of regulatory enterprise information governance: It is neither an end point nor an actual business architecture or in itself a framework.

It can be inferred that the definition here will to some extent mirror the Information Governance model, in that there will be model structures relating to the disposition of framework, policies, standards and so on. For an IT TOM such as a CBM-BoIT [157] ("Business of IT" Component Business Model™), we could expect that these elements and competences should be brought out and by a designer with closer application to the scope of Information Technology operations, per se, rather than to the governance of enterprise information.

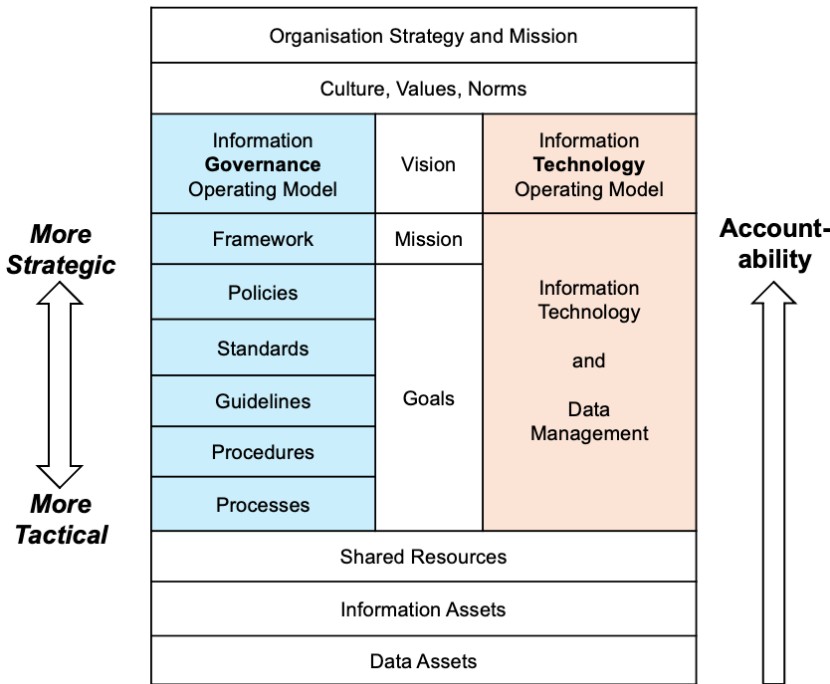

**Figure 1:** A graphical visualization of the novel definition of "Enterprise Information Governance" versus Information Technology, against vision, mission and goals in the context of organizational strategy and mission to support understanding of the new definition.

This definition might now, reasonably, and usefully be synthesized further and summarized without losing essence and meaning determined, as: "Enterprise information governance is a strategic framework that acts through control mechanisms designed to assure accountability in managing decision rights over data assets. It aligns day-to-day data and tactical operations with the organization's vision, in pursuit of value-maximization."

In the figure shown, "Information Governance Operating Model" represents Luisi's "business process models, product hierarchies, and business capability models, all with their corresponding taxonomies and business definition"[158] brought into the new definition. It is the strategic overview that incorporates the framework that brings together other governance components. It is this that determines the "business capabilities, functions, processes, organizational units, knowledge, information and trademarks that the organization will need... [and] the main characteristics of these elements and their desired interrelationships"[159] in a defined TOM.

"Framework" [160, 161, 162, 163] represents the operating model, or target operating model for information governance: The "cross-functional framework for managing data as a strategic enterprise asset"[164] and the componentry of the business architecture that determines actionable descriptions of the elements. It is this that performs the exercising of "authority, control, and accountability in managing decision rights over data" that we met previously. We can say that a framework in this context describes the overarching view of components and approaches to deliver enterprise information governance.

Below this level, we see "Shared Resources" within and across the organization, and then the "information assets" and "data assets" represented. Shared resources can represent

people, roles, facilities, but also data resources such as operational data stores (databases, data warehouses, marts, lakes, lakehouses, Amazon S3, Azure Blob storage), data fabrics, ETL and integration tooling, data catalogues, data dictionaries, and more.

Regulatory governance will need to encompass more technical topics, including aspects of personal data management, data security, data privacy and topics that intersect with CISO, DPO and CPO, policies and standards will be defined that specify the relevant regulations appropriately.

The novel definition's concept of a framework will encompass organizational components, business capabilities, and functions, the target business architecture of Hadaya and Gagnon [165]; the TOM of Campbell [166] and Anger [167]; the business capability models of Luisi [168] that direct and manage governance and define Weber, Otto and Österle's roles [169] —we met candidate roles of Owner (or "data trustee"), Custodian and Steward earlier, for example- and including those of strategic direction, advocacy, sponsorship and the executive role, "Executive Sponsor", that provides oversight, owns Lockwood's end-to-end vision, [170] and oversees alignment with Brous' data management organization and the tactical operational policies [171] that will appear as (for example) Gartner's, Lichtenberger's, Khatri and Brown's, and Smallwood's "Policies" and "Standards" [172, 173, 174, 175, 176]. Susan deMaine's observation that, "Experts largely agree that information governance requires buy-in, and preferably direction, from the top of the organization"[177] underlines the need for overarching, cross-organizational leadership in governance.

## 6. Conclusion and Future Research

This paper has synthesized a new definition of Enterprise Information Governance. Definitions of such governance in the literature to date have seen such governance clearly neither as a business architecture nor as an operating model: This paper has taken the perspectives of both the practitioner and of the scholar to do so in the context of enterprise architectures and to offer a different perspective to reframe such governance. This enables data strategists, governance designers, enterprise and business architects, scholars and researchers to use the definition as a basis for their work with greater clarity as business architectures for governance.

The paper has built upon earlier definitions to take a novel and more clearly regulatory approach and to synthesize the new definition for such governance; to build out a view of it as a scalable regulatory framework for large or complex organizations that sees information governance as a business architecture in the context of enterprise architecture; and to represent the definition visually to support understanding, assessments, novel developments and designs.

In Susan deMaine's thinking, we saw her finding that information governance is holistic. Robert Seiner and Olivia Benfeldt Nielsen and Boris Otto, Vijay Khatri and Carol Brown, Lockwood, et al., Pierce, et al., deMaine, and Atlan, amongst others, relating governance to assets; the British Academy and Royal Society paper as considering mechanisms to be within institutions; Steve Lockwood, as using technology to sustain and grow a business; Boisot and Canals' consideration of information in the physical world; Michael Buckland's information-as-thing; IBM's data objects; and Michael Madison's flowing and form-like data.

In the corporate world, the term "governance" appears often in the management of companies and boards of directors as corporate governance [178]; or, more broadly, of institutions as what could be referred to as institutional governance. Robert Smallwood notes

that "IG programs are driven from the top down but implemented from the bottom up,"[179] and The Sedona Conference® (quoted by Smallwood) determines that, "An [Information Governance] program should maintain sufficient independence from any particular department or division to ensure that decisions are made for the benefit of the overall organization."[180]

Future work should consider the definition of one or more comprehensive roadmaps for devising integrated enterprise information governance frameworks (strategies) that align closely with broader organizational goals in enterprises and will consider how to formulate novel, pragmatic candidate governance business architectures or target operating models (TOMs) that will define all the elements needed to translate a formulated data strategy (or, as this paper now makes clearer, information governance strategy) or initiative into the operational business of an organization and to achieve the desired target state [181].

We have seen thoughts from Nisha Talagala and Michael Madison earlier in this paper, but what of Humby's "New Oil" more recently? In early 2024, Christine Ashton and Sue Forder have a new perspective. They say that "An oil comparison oversimplifies data's pervasive nature."[182] Whilst "Data is an organization's most valuable asset, which hasn't changed," their strong case is that data today is less like oil, and more like yellowcake uranium: Relatively safe until refined, but then a potential "destroyer of worlds."

## 7. **Research Method**

Research for this paper was scoped around the determination of aspects of the regulatory governance of information and data in the context of information technology, enterprise architecture, and information technology architecture. Research was undertaken via a literature review, conducted at Oxford University's Bodleian Library, using search terms including "Enterprise Information Governance", "Information Governance", and "Data Governance", and the SOLO (Search Oxford Libraries Online) electronic search facility. This was supplemented by use of Elsevier's Scopus database; O'Reilly's Learning Platform (oreilly.com/library); ITHAKA's JSTOR journal articles, books, and images database; and Clarivate's ProQuest database of scholarly journals, books, dissertations and theses.

Methodologically, this was qualitative research with data collection and data analysis undertaken iteratively against an evolving list of references generated by the research, with a scope boundary determined in this case by time and by a scholarly assessment of relative thematic saturation within the time boundary: Generation of terms sufficient such that additional terms would add no apparent further insightful value beyond the time boundary determined by the researcher. A synthetic approach was adopted that would consider the texts determined to be within scope and formulate an analytical tabulation to review content by topic and theme. The themes, or categories being generated and work progressing following Creswell's "collection of data... data analysis that is both inductive and deductive and establishes patterns or themes" and including "the reflexivity of the researcher, a complex description or interpretation of the problem and its contribution to the literature." [183] As the review of sources unfolded, themes were added to research notes with quotations recorded in them to form a thematic concordance and reflexive aspects included. This was supplemented with approved access to view historic company texts and materials made available with the support of IBM's Data Services consulting practice in London. These were included in research by relative assessment of relevance within theme and topic.

## Conflicts of Interest

The author has previously been an IBM Corporation staff member and is a Chartered Fellow of the British Computer Society and a Fellow of the Institution of Engineering and Technology. The author is unaware of any conflicts of interest.

## Acknowledgements

The author is indebted to Dr Leon van Heerden, Paul Jarvis and Linnet Sen of the IBM Consulting Data Services practice in London, and to Dr Ian Dix and Dr Simon Bradford of AstraZeneca for their support and encouragement with respect to research for this paper.

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
