# OpenReview forum: "Defining a new perspective: Enterprise Information Governance"
_SEMANTiCS.cc/2024/Workshop/NXDG — NXDG 2024_

### Official Review · ~Melanie_Verstraete1 · 2024-07-26
**An interesting topic with potential for discussion, but it is unclear how the actual workshop will be structured.**

**Rating:** 6
**Confidence:** 3

**Review:**

The paper discusses a highly relevant topic concerning the definition of enterprise information governance and related terms. However, there is no clear mention of methodology, how the framework (visualization) is constructed, or how the papers are collected (e.g. structured literature review, scopus review). The text feels as a summary or literature review but it is unclear how this will be transformed into a workshop (also it is quite an abstract topic). Nevertheless, it does provide interesting discussion material as there is a lot of unclarity concerning governance, especially with the focus on data and information management and exchange.

The concept of information governance is only approached from the individual firm's perspective (enterprise perspective thus), but what about other perspectives such as data ecosystems? Providing further background and context regarding governance concerning data and information exchange between firms could be beneficial to the paper and would also make it even more fitting for this conference.

---

### Official Review · ~Harshvardhan_J._Pandit1 · 2024-08-02
**Discusses recent requirements to rethink data governance but unclear about what exactly it improves upon existing literature**

**Rating:** 6
**Confidence:** 3

**Review:**

The work outlines a good overview of the different discussions surroudning data governance and also outlines the high level concepts that define the subject. This leads to an interesting discussion about what data governance needs to evolve in to next. The paper is written in the style of "reiterating first principles" - which is necessary as the term "data governance" has devolved into numerous ambigous terms and practices.

In terms of limitations - the work is unclear in terms of what exactly it aims to introduce or change in comparison with the literature. For example, Figure 1 summarises the discussion into a 'framework' which includes the terms and processes used in enterprise domains to implement data governance. However there is a lack of detials for how these topics are currently being understood and if there is a new definition or direction for them (e.g. policies have many different theories and implementation). The figure also highlights that the work does not take into account existing knowledge to highilght what should change in practice e.g. see "Designing Data Governance" by Khatri and Brown which also provides a framework for data governance is often used as an educational resource to teach the subject.

Minor issues: the term "enterprise information governance" already has uses, so this isn't the first use of that phrase. This can come across as lack of awareness regarding the domain. The sections used to structure the paper represent the order of topics being discussed, but to facilitate this paper being applied the paper can be sectioned based on the topics highlighted in the figure with the contribution highlighted clearly with comparison to the literature. This will help people who are familiar with the subject to understand what is new/novel in this work.

---

### Decision · Program_Chairs · 2024-08-02

Accept